# Glycemic Variability Impacted by SGLT2 Inhibitors and GLP 1 Agonists in Patients with Diabetes Mellitus: A Systematic Review and Meta-Analysis

**DOI:** 10.3390/jcm10184078

**Published:** 2021-09-09

**Authors:** Heeyoung Lee, Se-eun Park, Eun-Young Kim

**Affiliations:** 1Department of Clinical Medicinal Sciences, Konyang University, Nonsan 32992, Korea; pharmdlee1@konyang.ac.kr; 2Department of Internal Medicine, Division of Endocrinology and Metabolism, Kangbuk Samsung Hospital, Sungkyunkwan University School of Medicine, Seoul 03063, Korea; seeun0630@gmail.com; 3Evidence-Based and Clinical Research Laboratory, Department of Health, Social and Clinical Pharmacy, College of Pharmacy, Chung-Ang University, Seoul 06974, Korea

**Keywords:** diabetes mellitus, GLP 1 agonist, glycemic variability, SGLT2 inhibitor

## Abstract

To investigate the effect of sodium-glucose cotransporter 2 (SGLT-2) inhibitors and glucagon-like peptide 1 (GLP-1) agonists on glycemic variability (GV), the mean amplitude of glucose excursion (MAGE), mean blood glucose (MBG) levels, and percentage of time maintaining euglycemia were evaluated. Randomized controlled trials evaluating the efficacy of SGLT-2 inhibitors and GLP-1 agonists for treating people with diabetes were selected through searches of PubMed, EMBASE, and other databases. Sixteen studies were finally analyzed. There were no differences in the reductions in MAGE after treatment with SGLT-2 inhibitors or GLP-1 agonists (standardized mean difference (SMD) = −0.59, 95% CI = −0.82 to −0.36 vs. SMD = −0.43, 95% CI = −0.51 to −0.35, respectively), and treatment with SGLT-2 inhibitors was associated with an increased reduction in MBG levels (SMD = −0.56, 95% CI = −0.65 to −0.48, *p* < 0.00001). Monotherapy and add-on therapy with medications were correlated with MAGE and MBG level reductions. In conclusion, SGLT-2 inhibitors and GLP-1 agonists were associated with a reduction in GV and could be alternatives for treating people with diabetes.

## 1. Introduction

Glycemic control is an important concern in diabetes care and associated with a reduced risk of macrovascular and microvascular complications [1]. To avoid vascular complications and improve glycemic control, glycemic variability (GV), which refers to glycemic fluctuation, is considered a clinical predictor and an important target in the treatment of diabetes [2,3]. Although glycated hemoglobin (HbA1c) has historically been the reference parameter indicating the risk of complications during the treatment of diabetes, GV is unlikely to be appropriately correlated with HbA1c levels [3]. The International Consensus on Use of Continuous Glucose Monitoring recently incorporated a coefficient of variation of <36% as a key indicator of primary GV to define stable diabetes [4]. As the use of a continuous glucose monitoring system (CGMS) is recommended to assess diabetes treatment, indexes representing GV, such as the mean amplitude of glucose excursion (MAGE), mean blood glucose (MBG) levels, and percentage of time maintaining euglycemia should be extensively evaluated [3,5] to verify the tradeoffs in glycemic targets [6].

Emerging evidence suggests that sodium-glucose cotransporter 2 (SGLT-2) inhibitors and glucagon-like peptide 1 (GLP-1) agonists effectively achieve glycemic control [1,7]. Although SGLT-2 inhibitors and GLP-1 agonists show pharmacological differences in mechanism of action (inhibiting the reabsorption of renal glucose or stimulating insulin secretion, respectively) [1,8], they effectively reduce glucose levels, cardiovascular complications, and even mortality in people with diabetes. Several head-to-head or network meta-analyses [1,7,8,9] indicated that SGLT-2 inhibitors and GLP-1 agonists effectively improve glycemic control and relative complications in clinical outcomes associated with HbA1c, such as cardiovascular and renal outcomes. However, to the best of our knowledge, no head-to-head meta-analyses have evaluated GV, including net changes in MAGE, MBG levels, and percentage of time maintaining euglycemia, during diabetes treatment with SGLT-2 inhibitors or GLP-1 agonists. This systematic review and meta-analysis aimed to investigate the effects of SGLT-2 inhibitors and GLP-1 agonists on GV measured using MAGE, MBG levels, and percentage of time maintaining euglycemia.

## 2. Materials and Methods

This systematic review and meta-analysis was conducted according to the Preferred Reporting Items for Systematic Reviews and Meta-Analysis statement (Appendix A) without registration of the study protocol.

### 2.1. Data Sources and Searches

The PubMed, EMBASE databases were searched for relevant studies published before 5 August 2020. The database search was conducted using the search keywords “glycemic variability”, “glycemic fluctuations”, “mean amplitude of glycemic excursions”, “mean blood glucose”, and “percentage of time maintaining euglycemia” combined with “diabetes mellitus”, “glucagon-like peptide 1”, and “sodium-glucose cotransporter 2” along with relevant Medical Subject Headings (MeSH) terms and marketed names of GLP-1 agonists and SGLT-2 inhibitors. The search strategy targeted published articles that evaluated the effects of GLP-1 agonists and SGLT-2 inhibitors for treating diabetes and was limited to full-text articles written in English. The references of the collected articles and systematic reviews were manually searched to identify additional studies. Disagreements between investigators were resolved through discussion.

### 2.2. Study Selection

The titles and abstracts of the retrieved articles were evaluated by two independent investigators to isolate potentially relevant articles. All randomized controlled trials (RCTs) that enrolled patients with diabetes were selected. For inclusion, study treatment periods were required to be one week or longer, and treatments had to include GLP-1 agonists and SGLT-2 inhibitors. Studies providing outcomes with MAGE, as proposed by Service et al. [10], MBG levels, and percentage of time maintaining euglycemia assessed using a CGMS were included in the analysis. Animal studies, those with a sample size of fewer than five patients, those that enrolled nondiabetics, and articles written in languages other than English were excluded. Stand-alone published abstracts were excluded.

### 2.3. Data Extraction and Quality Assessment

Data extracted from the retrieved articles included publication year, therapy type, diabetes type, GLP-1 agonist or SGLT-2 inhibitor type, comparison type, sample size, age of the study population, HbA1c levels, percentage of time maintaining euglycemia (≥70 to ≤180 mg/dL), MBG levels, and MAGE.

Net changes in the MAGE, MBG levels, and percentage of time maintaining euglycemia were quantified as discrepancies between pre- and post-treatment measures. Data extraction and the assessment of internal study validity and quality were performed by two investigators. The risk of bias assessment tool developed by the Cochrane Collaboration was used to assess RCT quality [11]. Confidence levels were determined by evaluating the effect estimates for each outcome. Using the Grading of Recommendations, Assessment, Development, and Evaluation (GRADE) approach, evidence quality was evaluated as high, moderate, low, or very low based on the study limitations, inconsistency, indirectness, imprecision, and publication bias [12].

### 2.4. Data Synthesis and Analysis

The primary outcomes were differences in MAGE and MBG levels pre- vs. post-treatment with GLP-1 agonists or SGLT-2 inhibitors (intervention group) and non-treatment with these medications (comparator group) according to therapy types. Therapy types were divided into GLP-1 agonists or SGLT-2 inhibitors. Secondary outcomes included the evaluation of differences in MAGE and MBG levels between the intervention and comparator groups according to monotherapy or add-on therapy and type 1 diabetes or type 2 diabetes. We separately evaluated MAGE and MBG as placebo-controlled and active comparator. We also determined the overall changes in the percentage of time maintaining euglycemia before vs. after treatment. To evaluate MAGE and MBG using a fixed mono-dosage of SGLT-2 inhibitors, an additional analysis was performed of maximum approved dosage of SGLT-2 inhibitors. The overall effect size for the studies, expressed as the standardized mean difference (SMD) and 95% confidence interval (CI), were calculated using meta-analysis software. Statistical significance was set at *p* < 0.05.

I^2^ statistics were applied to determine the significance of heterogeneity among studies classified as low (<25%), moderate (25–50%), or high (>50%). Publication bias was assessed using funnel plots and Egger’s test [13]. The sensitivity analysis was completed by repeating the meta-analysis and replacing various results or values with those that were arbitrary or unclear, for instance, while assessing cases of heterogeneity. Meta-regression was used to examine the quantitative effects of the study details on effect size [14]. The meta-analysis was conducted using Review Manager (version 5.3; The Nordic Cochrane Center, The Cochrane Collaboration, Copenhagen, Denmark) and Comprehensive Meta-Analysis (version 3; Biostat Inc., Englewood, NJ, USA).

## 3. Results

### 3.1. Study Selection

A comprehensive search identified 131 potentially eligible articles from the PubMed and EMBASE databases. Initially, 75 and 56 eligible articles for GLP-1 agonists and SGLT-2 inhibitors, respectively, were identified. The full-text screen reduced the number of included articles to 51. Of those, 37 full-text articles were removed for the following seven reasons: (1) unmet inclusion criteria (*n* = 134), (2) insufficient outcomes for evaluation (*n* = 2), (3) findings published as a poster (*n* = 20), (4) study unfinished (*n* = 1). Thus, 15 studies were subjected to the quantitative and qualitative analysis. The manual search of the retrieved articles’ reference lists and other sources identified three more articles. Finally, 16 articles [15,16,17,18,19,20,21,22,23,24,25,26,27,28,29,30] were included in our analysis (Figure 1).

### 3.2. Study Description

The basic characteristics of the 16 included studies [15,16,17,18,19,20,21,22,23,24,25,26,27,28,29,30] are presented in Table 1. A total of 2799 patients were included in the analysis. SGLT-2 inhibitors were examined in 10 studies [15,16,18,19,20,21,25,27,29,30], whereas the effects of GLP-1 agonists on MAGE, MBG levels, and percentage of time maintaining euglycemia among patients with diabetes were evaluated in six studies [17,22,23,24,26,28]. Except for two studies [18,28], all studies included in the analysis involved the addition of GLP-1 agonists or SGLT-2 inhibitors to other antihyperglycemic medications to treat diabetes. Six studies [15,16,19,20,22,25] enrolled patients with type 1 diabetes, while the others included patients with type 2 diabetes in the examination of changes in MAGE and MBG levels, or percentage of time maintaining euglycemia. Fifteen studies described the net changes in MAGE, while five studies [19,24,27,29,30] did not provide MBG outcomes to enable the calculation of the net changes between pre- and post-treatment. Two studies [16,27] provided values for the percentage of time maintaining euglycemia, which made the specific analysis possible in the present study. The baseline characteristics of body mass index (BMI) and age are provided in Appendix A.

### 3.3. Primary Outcomes

There were no significant discrepancies between the efficacy of GLP-1 agonists and SGLT-2 inhibitors for reducing MAGE among patients with diabetes (SMD = −0.59, 95% CI = −0.82 to −0.36 vs. SMD = −0.43, 95% CI = −0.51 to −0.35, respectively; Figure 2a). Treatment with SGLT-2 inhibitors more effectively reduced MBG levels (SMD = −0.56, 95% CI = −0.65 to −0.48, *p* < 0.00001; Figure 2b) than treatment with GLP-1 agonists (SMD = −0.22, 95% CI = −0.51 to 0.07, *p* = 0.13).

### 3.4. Secondary Outcomes

In the current analysis, both monotherapy and add-on therapy with SGLT-2 inhibitors or GLP-1 agonists were associated with reductions in MAGE (*p* < 0.05) without significant differences in efficacy (I^2^ = 0%, *p* = 0.33; Figure 3a). Both monotherapy and add-on therapy were correlated with a reduction in MBG levels without significant differences in efficacy (Figure 3b). In addition, MAGE was not significantly different between type 1 and type 2 diabetic treatments with SGLT-2 inhibitors or GLP-1 agonists (I^2^ = 0%, *p* = 0.65; Figure 4a). For MBG, SGLT-2 inhibitors or GLP-1 agonists contributed more to the reduction in type 1 diabetes than in type 2 diabetes (SMD = −0.56, 95% CI = −0.65 to −0.47 vs. SMD = −0.28, 95% CI = −0.55 to −0.01; Figure 4b). SGLT-2 inhibitors were better able to reduce MAGE when compared to placebo-controlled rather than to active-comparator (I^2^ = 78.9%, *p* = 0.03; Figure 5a). For MBG, comparator type did not differ between GLP-1 agonists and SGLT-2 inhibitors (Figure 5c,d). The percentage of time maintaining euglycemia was significantly higher in the intervention group than in the comparator group (SMD = 0.68, 95% CI = 0.32–1.03, *p* = 0.0002) without heterogeneity (Figure 6). Although no difference in MAGE change was observed between GLP-1 agonists and SGLT-2 inhibitors using fixed mono-dosage (I^2^ = 22.7%, *p* = 0.26; Figure 7a), the MBG reduction was significantly associated with a fixed mono-dosage of SGLT-2 inhibitor vs. GLP-1 agonists (SMD = −0.64, 95% CI = −0.76 to −0.52 vs. SMD = −0.22, 95% CI = −0.51 to 0.07, respectively; Figure 7b).

### 3.5. Risk of Bias and Level of Evidence

A summary of the risk of bias assessment is shown in Figure 8. The majority of the included studies showed a low risk of bias, and no cases of publication bias were identified (*p* = 0.520; Figure 9). Table 2 demonstrates the level of evidence assessed using the GRADE approach of the efficacy of the interventions for treating diabetes.

### 3.6. Meta-Regression Analysis

Baseline BMI (coefficient = −0.010; 95% CI = −0.077 to 0.057; *p* = 0.769; Appendix A) and age (coefficient = 0.008; 95% CI = −0.011 to 0.026; *p* = 0.414; Appendix A) did not significantly influence the effect of SGLT-2 inhibitors and GLP-1 agonists on MAGE reduction. However, BMI significantly influenced the MBG level reduction by SGLT-2 inhibitors and GLP-1 agonists (coefficient = −0.075; 95% CI = −0.149 to −0.001; *p* = 0.048, Appendix A), whereas age did not (coefficient = 0.001; 95% CI = −0.024 to −0.027; *p* = 0.912, Appendix A).

## 4. Discussion

This systematic review and meta-analysis evaluated the efficacy of SGLT-2 inhibitors and GLP-1 agonists for improving GV in diabetes treatment. SGLT-2 inhibitors and GLP-1 agonists were both significantly associated with reductions in MAGE and MBG levels and increases in the percentage of time maintaining euglycemia in patients with diabetes. Maintaining normoglycemia or near-normoglycemia is a pivotal goal for preventing or minimizing acute and chronic complications in diabetes care [2,31]. Considering the deleterious effects of intermittent hyperglycemia [3] and the relationship between a higher GV and the risk of developing hypoglycemia, a reduction in GV may be an important target for glycemic control using glucose-lowering therapy [2,31]. Mechanistic evidence has also shown that apoptosis is amplified in human umbilical vein endothelial cells exposed to alternating 5 and 20 mmol/L glucose levels, a phenomenon that is more deleterious than consistently high glucose levels [32]. Since HbA1c, the standard indicator of glycemic control, incompletely expresses GV [3], other parameters such as MAGE, MBG level, and percentage of maintaining euglycemia reflect GV [3]. MAGE quantifies major swings in glycemia, excludes minor swings, refines the characterization of glycemic fluctuations, and represents the mean change between consecutive peaks and nadir values exceeding one standard deviation around the mean 24 h glucose value [3]. As it plays a key role in the pathogenesis of diabetes complications caused by major swings in glycemia, a reduction in MAGE after treatment with glucose-lowering medications may be a useful independent predictor of a decrease in the risk of cardiovascular events [33]. Furthermore, since a higher MBG level correlates with a higher GV [3], reductions in net changes in MAGE or MBG levels and increases in the time maintaining euglycemia between pre- and post-treatment with SGLT-2 inhibitors and GLP-1 agonists, which reflect a lower GV, as described herein, represent glycemic control in patients with diabetes and can possibly predict a reduction in the risk of adverse cardiovascular events.

SGLT-2 inhibitors and GLP-1 agonists are differently associated with reductions in MAGE and MBG levels according to the type of therapy in the present analysis. Treatment with SGLT-2 inhibitors and GLP-1 agonists is associated with reductions in MAGE in patients with diabetes. However, treatment with SGLT-2 inhibitors is more significantly correlated with reductions in MBG levels than treatment with GLP-1 agonists. Although Patoulias et al. [8] reported that GLP-1 agonists are more efficacious than SGLT-2 inhibitors at improving glycemic control and HbA1c levels in patients with type 2 diabetes (mean difference = −0.38; 95% CI = −0.55, −0.22), the small number of studies included in that analysis limited the significance of those results. In contrast, a previous network meta-analysis showed that SGLT-2 inhibitors were superior to short-acting GLP-1 agonists at reducing HbA1c levels [9]. Except for one study, all studies used short-acting GLP-1 agonists and considered the close association between HbA1c and MBG levels [5], indicating that treatment with SGLT-2 inhibitors may be more closely related to a reduction in MBG levels consistent with the results of a previous study [9]. A greater reduction in MBG levels in patients with diabetes following treatment with SGLT-2 inhibitors vs. GLP-1 agonists may also be explained by the difference in their mechanisms of action. Pharmacologically, SGLT-2 inhibitors function by inhibiting renal glucose reabsorption, an insulin-independent mechanism for lowering blood glucose levels [34], whereas GLP-1 agonists act by increasing glucose-dependent insulin secretion, constraining glucagon secretion, and slowing gastric emptying [35]. Considering the discrepancies in efficacy between SGLT-2 inhibitors and GLP-1 agonists caused by mechanistic differences [34,35] or GLP-1 agonist type [9], in practice, SGLT-2 inhibitors and GLP-1 agonists should be cautiously applied to reduce MBG levels during the treatment of diabetes.

Despite limited clinical trials on SGLT-2 inhibitors or GLP-1 agonist monotherapy, these therapeutic agents have been investigated as new classes of antidiabetic drugs and are recommended as add-on therapies for treating patients with diabetes [36,37]. Our study failed to find differences between monotherapy and add-on therapy with SGLT-2 inhibitors or GLP-1 agonists for reducing MAGE and MBG levels in patients with diabetes. SGLT-2 inhibitors and GLP-1 agonists have the ability to improve cardiovascular outcomes and/or renal outcomes [1,7,8,9,38]; however, more studies are required to evaluate their various effects, including those on GV.

Although SGLT-2 inhibitors and GLP-1 agonists were more effective against GV fluctuations than other drugs, certain aspects must be elucidated. A meta-analysis revealed that dipeptidyl peptidase IV inhibitor (DPP-4) inhibitors were significantly more effective than other oral antihyperglycemic drugs at reducing GV in patients with type 2 diabetes [39]. However, the effects of DPP-4 inhibitors on cardiovascular outcomes are considered neutral [39,40,41]. This could lead to reasonable doubt about the effect of GV on cardiovascular outcomes. While several studies have compared the effectiveness and safety of SGLT-2 inhibitors or GLP-1 agonists to those of DPP-4 inhibitors, only a few have directly compared the effects of GV [1,7,8,9]. Only two reports in the current study compared dapagliflozin to DPP-4 inhibitors [21,29]; however, in one study, the therapies were not administered simultaneously, while the second study was an open-label blinded end-point trial conducted for 12 weeks. Moreover, bias related to funding issues was rarely excluded from the evaluated reports.

Nevertheless, the possibility of a quantitative effect difference between GLP-1 agonists and DPP-4 inhibitors on GV is suspected. An additional subgroup analysis revealed that the effect size of a reduction in MAGE levels in patients with diabetes treated with GLP-1 agonists as add-on therapy was −0.60 (SMD; 95% CI = −0.88, −0.33; *p* < 0.00001). In a previous meta-analysis, the effect size of a reduction in MAGE with DPP4-inhibitors as add-on therapy was −0.48 (SMD; 95% CI = −0.84 to −0.12; *p* = 0.008) [40]. Although the effect size of GLP-1 agonists is larger than that of DPP4 inhibitors, the CI ranges overlapped. Hopefully, future studies on the influence of GV on diabetic complications in response to treatment with these drugs can increase our understanding of the role of GV in the development of related complications.

The current study has some limitations. First, we did not investigate the cardiovascular outcomes related to our targeted indexes such as MAGE, MBG levels, and percentage of time maintaining euglycemia in the treatment with SGLT-2 inhibitors and GLP-1 agonists. Since the correlation between cardiovascular outcomes and our targeted values was beyond the scope of this study, further studies are required. Second, we did not evaluate the cost-effectiveness of SGLT-2 inhibitors and GLP-1 agonists for treating diabetes. Despite the need to evaluate the cost-effectiveness of treatments with glucose-lowering medications [36], the current study only evaluated GV following treatment with SGLT-2 inhibitors and GLP-1 agonists in diabetes care. Hence, cost-effectiveness should be evaluated in future studies. Third, only two studies that evaluated the effect of interventions on the percentage of time maintaining euglycemia were included in this analysis. This limited number of studies was unable to provide significant evidence to show the effects of SGLT-2 inhibitors and GLP-1 agonists on the percentage of time maintaining euglycemia in patients with diabetes. However, considering the importance of suggesting effective glycemic targets for monitoring the daily GV of glucose-lowering medications [42], the current study shows that an evaluation of the time to maintain euglycemia in diabetes care may be an alternative target for future diabetic therapy studies. In the future, updated data on the effect of “the time maintaining euglycemia” in diabetic complications may help explain the role of GV in the onset of these complications [2,3]. Fourth, only short-term GV was analyzed in the present study. Hence, in future studies, long-term GV should be determined based on per-visit measurements of HbA1c and fasting plasma glucose levels to enable the calculation of SD and coefficient of variation [3]. Fifth, the current study could not evaluate the different effects of monotherapy and add-on therapy in each RCT using different anti-diabetic drugs. Due to the limited number of studies using monotherapy, specific evaluations with different anti-diabetic drugs could not be performed. Finally, the present study focused on evaluating surrogate markers related to GV without examining safety issues, such as euglycemic diabetic ketoacidosis related to the treatment of diabetes. SGLT-2 inhibitors were recently reported to be associated with euglycemic diabetic ketoacidosis; however, only a limited number of well-structured clinical trials have provided supportive evidence [43]. Pathophysiologically, patients with diabetes showing euglycemia can develop ketoacidosis after treatment with SGLT-2 inhibitors; however, evaluating the safety of SGLT-2 inhibitors was beyond the scope of the present study. Therefore, more highly qualified clinical trials of SGLT-2 inhibitor safety in various settings are needed.

## 5. Conclusions

In conclusion, the efficacy of SGLT-2 inhibitors and GLP-1 agonists at lowering GV was higher than that of placebo or other therapies for the treatment of diabetes. Considering the importance of GV in cardiovascular events, the current study showed that SGLT-2 inhibitors and GLP-1 agonists are favorable alternatives for treating diabetes to prevent cardiovascular complications. Although patient-specific regimens should be applied in practical situations, the use of SGLT-2 inhibitors and GLP-1 agonists should be recommended to reduce GV in patients with diabetes.

## Figures and Tables

**Figure 1 jcm-10-04078-f001:**
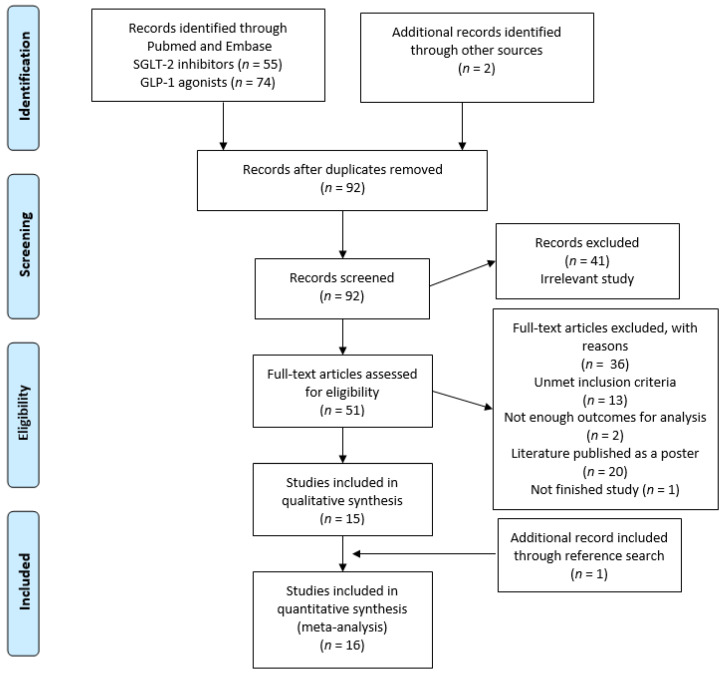
The flowchart of the study selection process for the meta-analysis. GLP-1: glucagon-like peptide 1; SGLT-2: sodium-glucose cotransporter-2.

**Figure 2 jcm-10-04078-f002:**
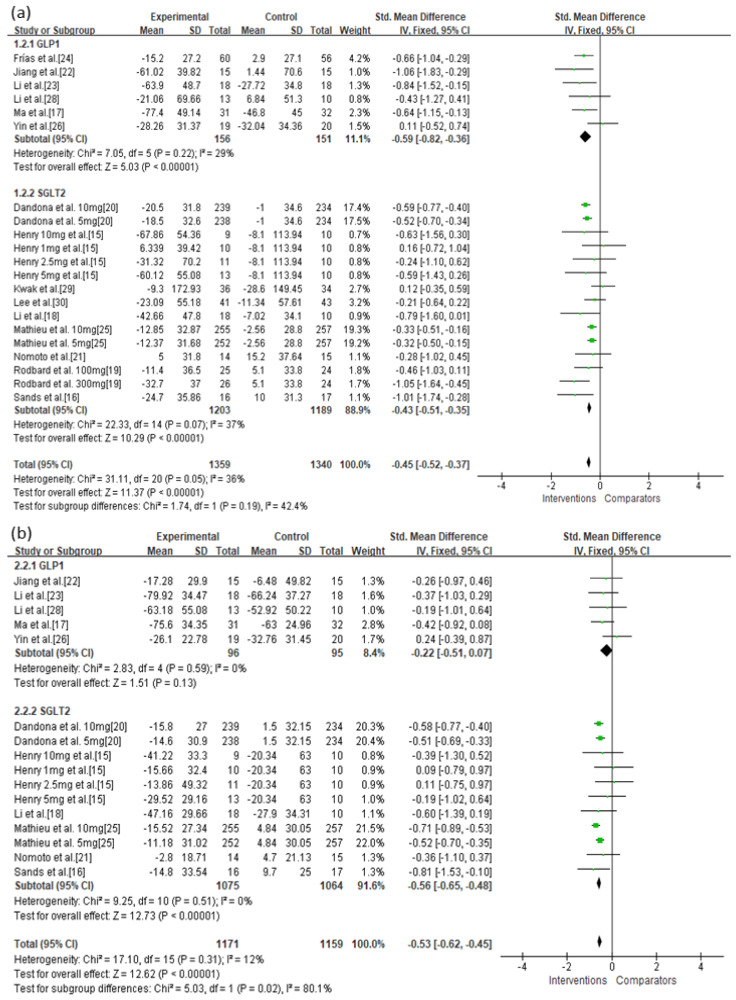
Effects of intervention on MAGE and MBG levels according to SGLT-2 inhibitors or GLP-1 agonists. (**a**) MAGE; (**b**) MBG level. MAGE, mean amplitude of glycemic excursion; MBG, mean blood glucose; GLP-1, glucagon-like peptide 1; SGLT-2, sodium-glucose cotransporter 2. Green squares suggested measure of effect for each of included studies, and black diamond represented as meta-analyzed measure of effect. Bold letters indicated a category or subtotal of each subgroup and overall outcome.

**Figure 3 jcm-10-04078-f003:**
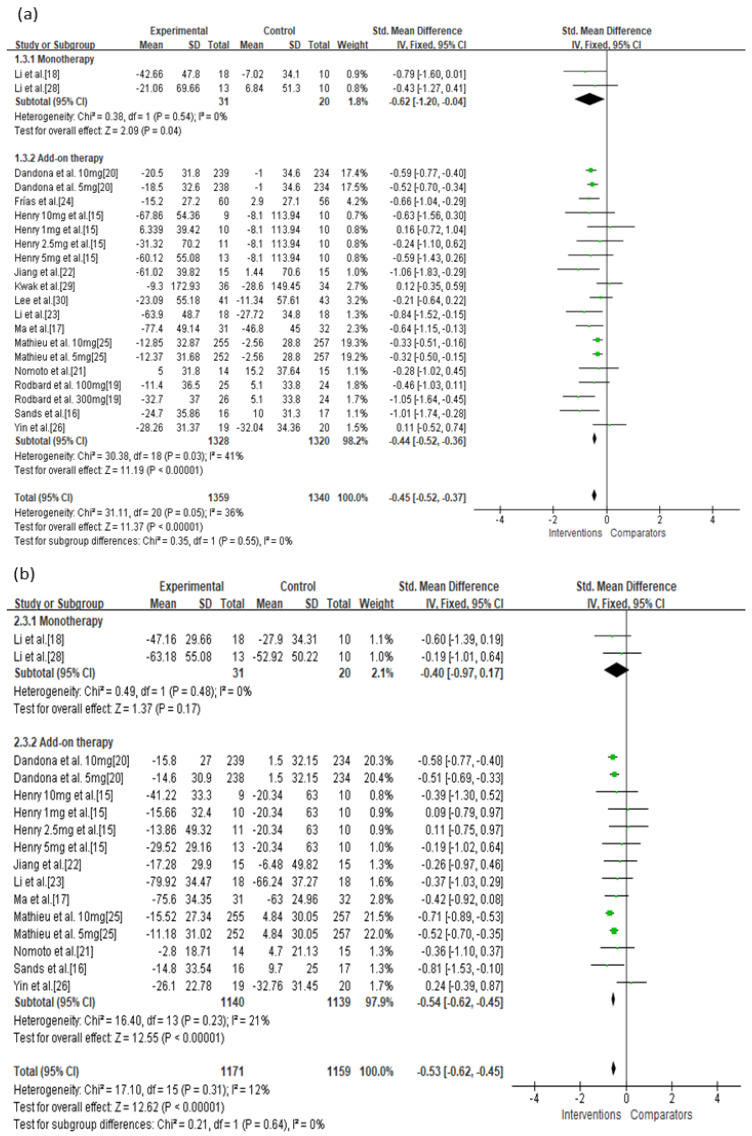
Effects of intervention on MAGE and MBG levels according to monotherapy or add-on therapy. (**a**) MAGE; (**b**) MBG level. MAGE, mean amplitude of glycemic excursion; MBG, mean blood glucose; GLP-1, glucagon-like peptide 1; SGLT-2, sodium-glucose cotransporter 2. Green squares suggested measure of effect for each of included studies, and black diamond represented as meta-analyzed measure of effect. Bold letters indicated a category or subtotal of each subgroup and overall outcome.

**Figure 4 jcm-10-04078-f004:**
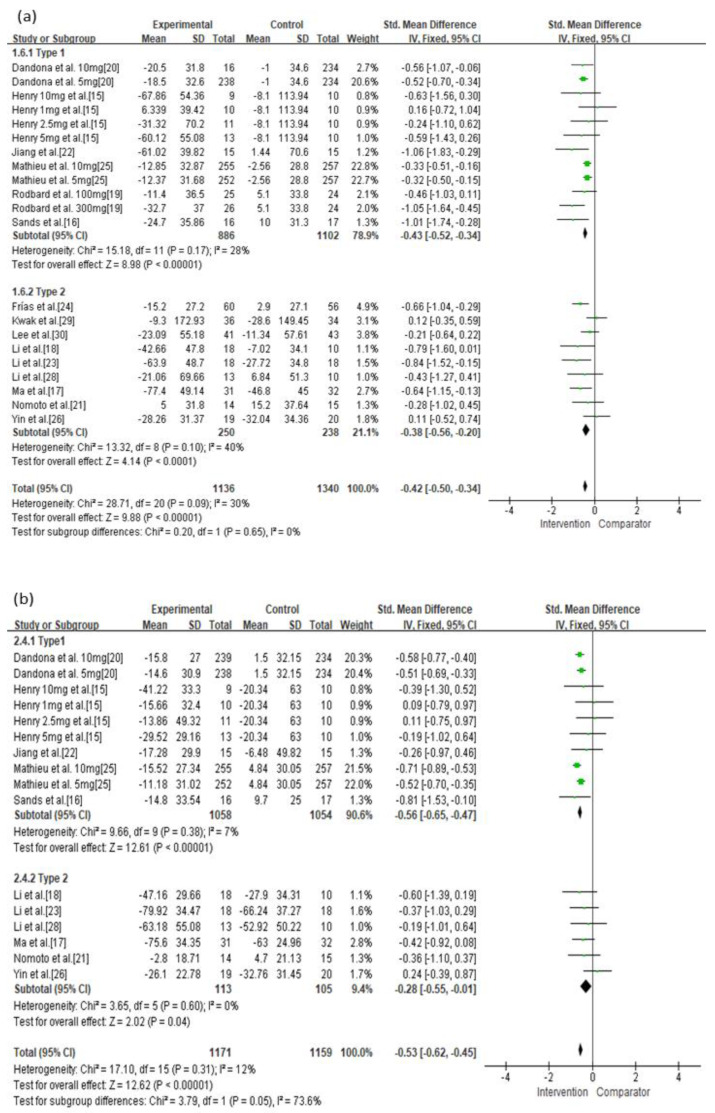
Effects of intervention on MAGE and MBG levels according to diabetes type. (**a**) MAGE; (**b**) MBG level. MAGE, mean amplitude of glycemic excursion; MBG, mean blood glucose; GLP-1, glucagon-like peptide 1; SGLT-2, sodium-glucose cotransporter 2. Green squares suggested measure of effect for each of included studies, and black diamond represented as meta-analyzed measure of effect. Bold letters indicated a category or subtotal of each subgroup and overall outcome.

**Figure 5 jcm-10-04078-f005:**
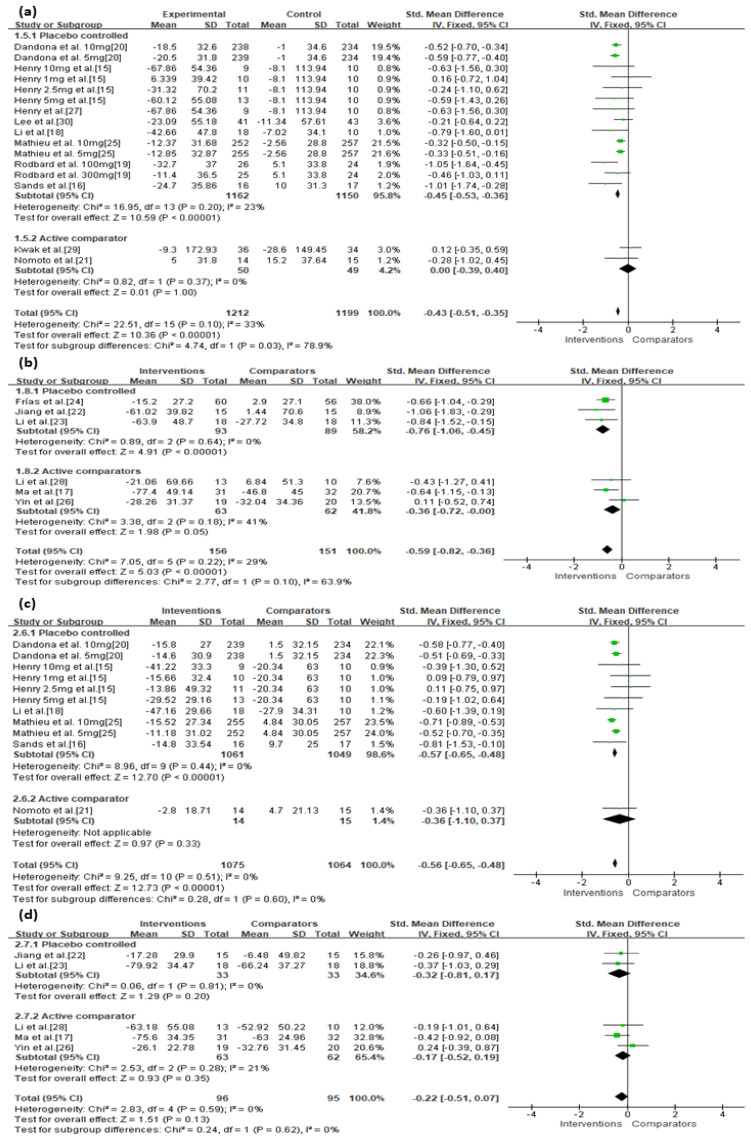
Effects of intervention on MAGE and MBG levels according to comparator type. (**a**) Evaluating MAGE with SGLT-2 inhibitor use; (**b**) evaluating MAGE with GLP-1 agonist use; (**c**) evaluating MBG with SGLT-2 inhibitors; (**d**) evaluating MBG with GLP-1 agonists. MAGE, mean amplitude of glycemic excursion; MBG, mean blood glucose; GLP-1, glucagon-like peptide 1; SGLT-2, sodium-glucose cotransporter 2. Green squares suggested measure of effect for each of included studies, and black diamond represented as meta-analyzed measure of effect. Bold letters indicated a category or subtotal of each subgroup and overall outcome.

**Figure 6 jcm-10-04078-f006:**
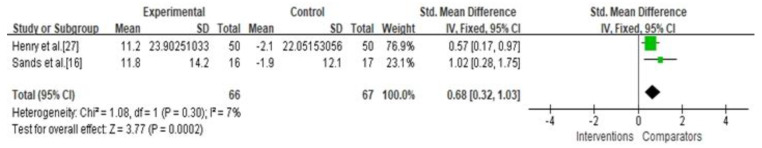
Effects of SGLT-2 inhibitors and GLP-1 agonists on the percentage of time maintaining euglycemia. CI, confidence interval; GLP-1, glucagon-like peptide 1; SGLT-2, sodium-glucose cotransporter 2. Green squares suggested measure of effect for each of included studies, and black diamond represented as meta-analyzed measure of effect. Bold letters indicated overall outcome.

**Figure 7 jcm-10-04078-f007:**
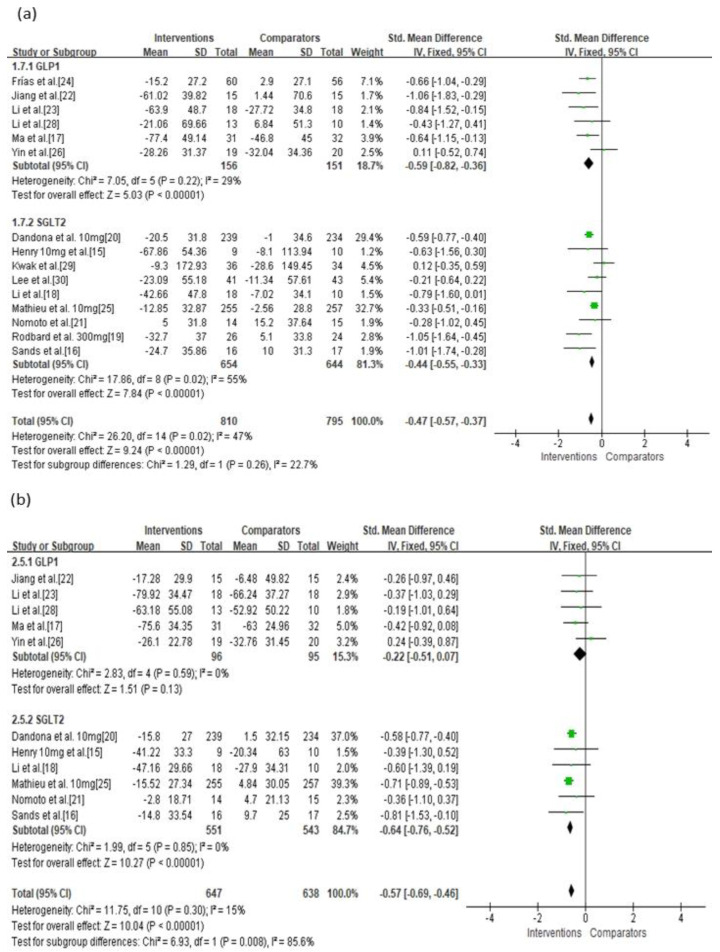
Effects of intervention on MAGE and MBG levels according to fixed mono-dosage of SGLT-2 inhibitors or GLP-1 agonists. (**a**) MAGE; (**b**) MBG level. MAGE, mean amplitude of glycemic excursion; MBG, mean blood glucose; GLP-1, glucagon-like peptide 1; SGLT-2, sodium-glucose cotransporter 2. Green squares suggested measure of effect for each of included studies, and black diamond represented as meta-analyzed measure of effect. Bold letters indicated a category or subtotal of each subgroup and overall outcome.

**Figure 8 jcm-10-04078-f008:**
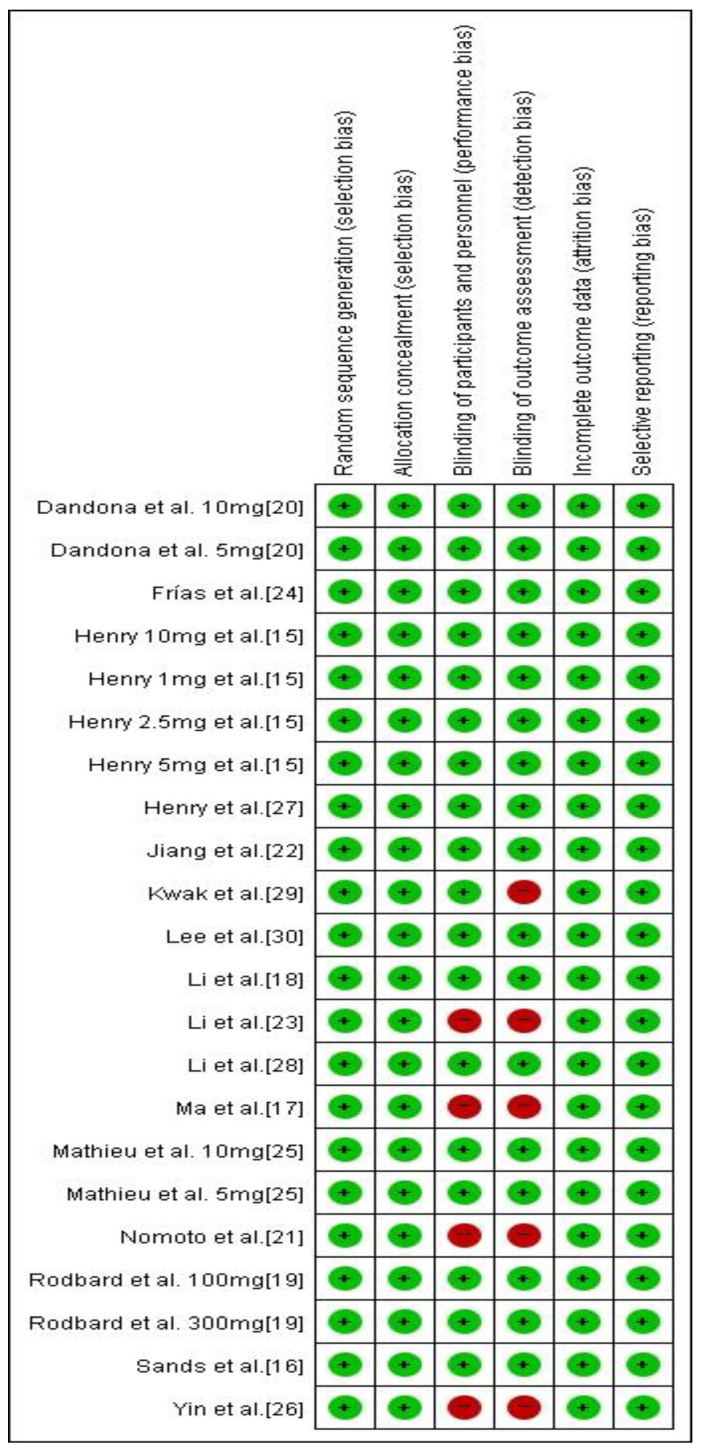
Risk of bias assessment of the studies included in the analysis.

**Figure 9 jcm-10-04078-f009:**
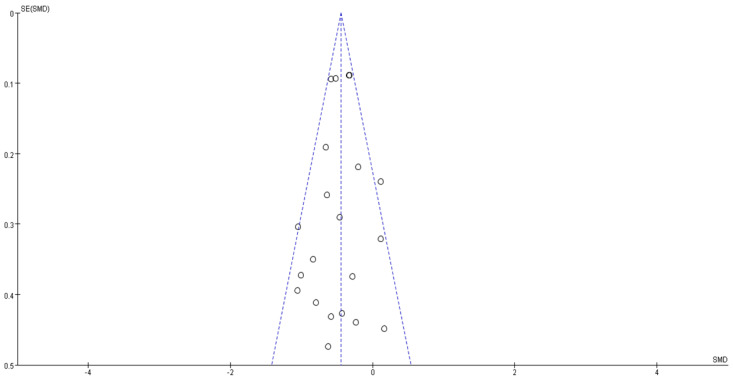
Funnel plot of publication bias. Open circles suggested included studies and diamond explained observed effect size.

**Table 1 jcm-10-04078-t001:** Characteristics of studies included in the analysis.

Study Name	Publication Year	Therapy Regimen	Diabetes Type	Intervention	Comparator
Henry et al. [15]	2015	Add-on therapy	Type 1	Dapagliflozin and insulin	Placebo and insulin
Sands et al. [16]	2015	Add-on therapy	Type 1	Sotagliflozin and insulin	Placebo and insulin
Ma et al. [17]	2015	Add-on therapy	Type 2	Liraglutide and metformin	Insulin and metformin
Li et al. [18]	2016	Monotherapy	Type 2	Dapagliflozin	Placebo
Rodbard et al. [19]	2017	Add-on therapy	Type 1	Canagliflozin and insulin	Placebo and insulin
Dandona et al. [20]	2017	Add-on therapy	Type 1	Dapagliflozin and insulin	Placebo and insulin
Nomoto et al. [21]	2017	Add-on therapy	Type 2	Dapagliflozin and insulin	DPP-IV and insulin
Jiang et al. [22]	2017	Add-on therapy	Type 1	Exenatide and insulin	Placebo and insulin
Li et al. [23]	2017	Add-on therapy	Type 2	Exenatide and insulin	Placebo and insulin
Frías et al. [24]	2017	Add-on therapy	Type 2	Exenatide and metformin	Placebo and metformin
Mathieu et al. [25]	2018	Add-on therapy	Type 1	Dapagliflozin and insulin	Placebo and insulin
Yin et al. [26]	2018	Add-on therapy	Type 2	Exenatide and metformin	Insulin and metformin
Henry et al. [27]	2018	Add-on therapy	Type 2	Dapagliflozin and insulin or metformin	Placebo and insulin or metformin
Li et al. [28]	2019	Monotherapy	Type 2	Dulaglutide	Glimepiride
Kwak et al. [29]	2020	Add-on therapy	Type 2	Dapagliflozin and metformin	Gemigliptin and metformin
Lee et al. [30]	2020	Add-on therapy	Type 2	Dapagliflozin and insulin and/or OADs	Placebo and insulin and/or OADs

DPP-IV, dipeptidyl peptidase IV inhibitor; OAD, oral antihypoglycemic drug; Type 1, type 1 diabetes; Type 2, type 2 diabetes.

**Table 2 jcm-10-04078-t002:** Summary of findings for MAGE and MBG level reduction upon comparing interventions to comparators based on the GRADE approach.

Outcome	Limitation	Inconsistency	Indirection	Imprecision	Publication Bias	Standard Mean Difference	Quality of Evidence
MAGE							
GLP-1	Serious	Not serious	Not serious	Not serious	Undetected	−0.59 (−0.82, −0.36)	⊕⊕⊕◯ Moderate
SGLT-2	Serious	Not serious	Not serious	Not serious	Undetected	−0.43 (−0.51, −0.35)	⊕⊕⊕◯ Moderate
Monotherapy	Not serious	Not serious	Not serious	Not serious	Undetected	−0.62 (−1.20, −0.04)	⊕⊕⊕⊕ High
Add-on therapy	Serious	Not serious	Not serious	Not serious	Undetected	−0.44 (−0.52, −0.36)	⊕⊕⊕◯ Moderate
Type 1	Not serious	Not serious	Not serious	Not serious	Undetected	−0.43 (−0.52, −0.34)	⊕⊕⊕⊕ High
Type 2	Serious	Not serious	Not serious	Not serious	Undetected	−0.38 (−0.56, −0.20)	⊕⊕⊕◯ Moderate
SGLT-2 vs. Placebo ^a^	Not serious	Not serious	Not serious	Not serious	Undetected	−0.45 (−0.53, −0.36)	⊕⊕⊕⊕ High
GLP-1 vs. Placebo ^b^	Serious	Not serious	Not serious	Not serious	Undetected	−0.76 (−1.06, −0.45)	⊕⊕⊕◯ Moderate
SGLT-2 vs. Active ^c^	Very serious	Not serious	Not serious	Not serious	Undetected	0.00 (−0.39, 0.40)	⊕⊕◯◯ Low
GLP-1 vs. Active ^d^	Serious	Not serious	Not serious	Not serious	Undetected	−0.36 (−0.72, −0.00)	⊕⊕⊕◯ Moderate
MBG							
GLP-1	Serious	Not serious	Not serious	Not serious	Undetected	−0.22 (−0.51, 0.07)	⊕⊕⊕◯ Moderate
SGLT-2	Serious	Not serious	Not serious	Not serious	Undetected	−0.56 (−0.65, −0.48)	⊕⊕⊕◯ Moderate
Monotherapy	Serious	Not serious	Not serious	Not serious	Undetected	−0.40 (−0.97, 0.17)	⊕⊕⊕◯ Moderate
Add-on therapy	Serious	Not serious	Not serious	Not serious	Undetected	−0.54 (−0.62, −0.45)	⊕⊕⊕◯ Moderate
Type 1	Not serious	Not serious	Not serious	Not serious	Undetected	−0.56 (−0.65, −0.47)	⊕⊕⊕⊕ High
Type 2	Serious	Not serious	Not serious	Not serious	Undetected	−0.28 (−0.55, −0.01)	⊕⊕⊕◯ Moderate
SGLT-2 vs. Placebo ^a^	Not serious	Not serious	Not serious	Not serious	Undetected	−0.57 (−0.65, −0.48)	⊕⊕⊕⊕ High
GLP-1 vs. Placebo ^b^	Serious	Not serious	Not serious	Not serious	Undetected	−0.32 (−0.81, 0.17)	⊕⊕⊕◯ Moderate
SGLT-2 vs. Active ^c^	Very serious	Not serious	Not serious	Not serious	Undetected	−0.36 (−1.10, 0.37)	⊕⊕◯◯ Low
GLP-1 vs. Active ^d^	Serious	Not serious	Not serious	Not serious	Undetected	−0.17 (−0.52, 0.19)	⊕⊕⊕◯ Moderate
Percentage of time maintaining euglycemia	Not serious	Not serious	Not serious	Not serious	Undetected	0.68 (0.32, 1.03)	⊕⊕⊕⊕ High

GLP-1, glucagon-like peptide 1 agonists; MAGE, mean amplitude of glycemic excursion; MBG, mean blood glucose; SGLT-2, sodium-glucose cotransporter 2 inhibitors; Type 1, type 1 diabetes; Type 2, type 2 diabetes; ⊕ = attainment of Grading of Recommendations, Assessment, Development, and Evaluation criteria; ◯ = uncertainty of attaining Grading of Recommendations, Assessment, Development, and Evaluation criteria; ^a^ comparing SGLT-2 inhibitors to placebo controlled; ^b^ comparing GLP-1 agonists to placebo controlled; ^c^ comparing SGLT-2 inhibitors to active comparator; ^d^ comparing GLP-1 agonists to active comparator.

## Data Availability

The data used in the current study are available upon request from the corresponding author.

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
