# Peer review of "Glycemic Variability Impacted by SGLT2 Inhibitors and GLP 1 Agonists in Patients with Diabetes Mellitus: A Systematic Review and Meta-Analysis"

_jcm, 2021, doi:10.3390/jcm10184078_

Round 1

Reviewer 1 Report

The authors review a very relevant subject namely the glycemic variability of SGLT-2i and GLP-1 agonists. The topic is relevant

I have some issues presented below.

Introduction

  • Something seems wrong with reference 4. It is missing []

Methods

  • Was this study registered on PROSPERO (https://www.crd.york.ac.uk/prospero/) ? if not please state so.
  • As a primary outcome the authors look at pooled data on patients with type 2 or type 1 diabetes in treatment with GLP-1 RA OR SGLT-2i. I can’t see any good reason for pooling results from studies on to different drug classes. I simply do not see what I as a reader should conclude from that?? What can I use that information for? In my opinion you should remove these results and focus on the secondary results presented

Results:

  • Section 3.1 page 3, line 124-127. Hard to follow, please rewrite. However, fig. 1 nicely picture the process.
  • Section 3.2 page 4 line 138, Did six studies included type 1 diabetes alone? Or both type 2 and type 2 please clarify. This finding is very surprising, because the neither GLP-1-RA or SGLT-2i is approved for use in patients with type 1 diabetes. I do know that this was not the aim of the study but maybe is should have been considered to ONLY examine type 1 or type 2. A mix is always confusion.
  • Fig 4. Something is wrong. It looks like you have put to figures on top of each other? I think it might be fig 5 that for some reason in now on top of Fig 4.a

Author Response

Please, confirm our responses on the attached file and changes of revised manuscript marked with "Track changes" function. 

Reviewer 2 Report

The authors investigated the effects of SGLT2i and GLP-1A on GV, MAGE, MBG levels, and the percentage of time maintaining euglycemia in patients with DM. They have conducted a review and meta-analysis by RCTs.

However, the article suffers from critical problems due to inappropriate study design. All results do not contribute to the thesis. The novelty of this article is poor.

Major comments:

1) In Table 1, and Fig. 2, Fig.3, The patients with T1DM and T2DM were mixed in the analysis. Since insulin secretion and etiologies are quite different in those, the authors should evaluate them individually. Can GLP-1A generally be used in T1DM?

2) In Table 1, and Fig. 2, Fig.3, In ‘intervention group’ and ‘competitor group’, for instance, ref 20 is a study regarding the comparison of SGLT2i and placebo. But ref 21 is a study regarding the comparison of SGLT2i and DPP-4i. Therefore, these studies cannot be mixed.

3) In Fig. 2, Fig. 3, Fig. 4, and Supplemental Table 1, for example, such as 5mg or 10mg of SGLT2i in the same RCT can be included in the whole analysis? The authors should consider the fixed mono-dosage throughout all SGLT2i as equivalent dosages, according to the original articles.

4) In Fig. 4a, a different figure was embedded in another figure? This figure is not understandable.

5) In Table 2, ‘Quality of evidence’ is arbitrary, and should be corrected.

6) Monotherapy or add-on therapy in each RCT using different anti-diabetic drugs should be analyzed individually.

7) It is interesting to compare the effects of SGLT2i, GLP-1A, DPP4-I, SU, Met, and glynide.

Minor comments:

1) ABSTRACT: In [MAGE after treatment with SGLT-2 inhibitors and GLP-1 agonists; however, treatment with SGLT-2 inhibitors], The word ‘however seems to be wrong.

2) ABSTRACT: ‘Conclusively’ to be ‘In conclusion’

3) Line 39 Please correct diabetes4 to diabetes [4]

4) Table 1 [16) should be corrected to [16].

5) Many English grammatical errors and typing errors are seen throughout the MS.

Author Response

(The authors gave the same response as above.)

Round 2

Reviewer 1 Report

I'm satisfied with the changes made by the authors and have not further issues.